# Magnitude of neonatal meningitis and associated factors among newborns with neonatal sepsis admitted to the University of Gondar Comprehensive Specialized Hospital, North Gondar, Ethiopia

**Mulugeta Nigusu Wondimu, Alemayehu Teklu Toni, Teshome Geletaw Zamanuel**⊙*

Department of Pediatrics and Child Health, College of Medicine and Health Sciences, University of Gondar, Gondar, Ethiopia

\* geletawteshome@yahoo.com

## Abstract

### Background

Neonatal meningitis continues to be a devastating infection with high mortality and morbidity worldwide. The prevalence of neonatal meningitis varies across countries. However, there is a paucity of studies on neonatal meningitis in most low-resource settings. Therefore, this study aimed to determine the magnitude, clinical presentations, and associated risk factors of meningitis among newborns with neonatal sepsis.

### Method

An institution-based, cross-sectional study was conducted among newborns with neonatal sepsis from May 1, 2021, to Oct 30, 2021. Neonates with neonatal sepsis admitted to the University of Gondar Comprehensive Specialized Hospital (UOGCSH) during the study period were included in the study. Single population proportion formula was used to calculate the sample size and a systematic random sampling technique was used to select the study participants. Two trained pediatric residents collected the data by using a pretested, structured questionnaire. The data was entered into Epi-info version 7.0 and exported to SPSS version 24.0 for analysis. Binary logistic regression was used to identify the associated factors. P-value<0.05 was considered statistically significant.

### Result

A total of 171 participants were enrolled in this study. The mean postnatal age was 10.74 ±8.0 days. The male to female ratio was l.3:1. The prevalence of neonatal meningitis among suspected sepsis was 19.3%; 95% CI [13.5%-25.1%]. The prevalence of meningitis was 22.8% in Early-Onset Neonatal Sepsis and 16.8% in Late-onset Neonatal Sepsis. Neurologic symptoms (seizure & altered mentation) were seen in 25% of the patients with meningitis. Two risk factors, Prolonged labor (adjusted odds ratio [AOR]: 4.98; 95% CI: 1.99–

**Data Availability Statement:** All relevant data are within the paper and its Supporting Information files.

**Funding:** The author(s) received no specific funding for this work.

**Competing interests:** The authors have declared that no competing interests exist.

12.48) and Prolonged Rupture of Membrane (AOR: 5:38; 95% CI: 1.92–14.42) were significantly associated with neonatal meningitis.

## Conclusion

The prevalence of neonatal meningitis was higher in early-onset neonatal sepsis than in late-onset neonatal sepsis. Obstetric factors were associated with neonatal meningitis. We recommend that routine CSF analysis should be practiced in all neonates with neonatal sepsis regardless of their postnatal age and clinical features. Early detection and treatment of obstetric factors may have the potential to reduce neonatal meningitis.

## Introduction

Neonatal meningitis continues to be a serious health challenge with high mortality and morbidity worldwid [1,2]. Neonatal meningitis is a diffuse inflammation of the meninges during the first 28 days of life [3]. Despite advances in the management of neonatal meningitis, mortality and neurologic complications are still unacceptably high, especially in developing countries [2,4]. Generally, newborns are uniquely prone to invasive diseases like neonatal sepsis, meningitis, and pneumonia due to their lack of fully responsive innate immunity [5,6]. Therefore, meningitis most commonly occurs in the first month of life than in any other period and the incidence ranges from 0.2–0.5 per 1000 live births, depending on the criteria used to define meningitis [7,8].

The prevalence of neonatal meningitis in newborns with bacteremia also varies ranging from 9 to 30% [6,9–13]. Previous studies stated that this prevalence differs in early-onset neonatal sepsis (EONS), [defined as an infection in the 1st week of age] and in late-onset neonatal sepsis (LONS), though the findings are contradicting [10,14]. A study conducted in Australia showed up to 10% with early-onset and nearly 25% of those with late-onset sepsis had neonatal meningitis [10,14]. In a study done in Mexico, the magnitude of neonatal meningitis with bacteremia was almost equal in both EONS and LONS (49% versus 51%) [15] whereas, in another study done in Iran, the prevalence was higher in EONS(65%) than in LONS(35%) [16].

The risk factors associated with neonatal meningitis in neonatal sepsis might vary from one study area to another [15,17–19]. Previous studies found that neonatal meningitis was high in neonates with prematurity [16], maternal history of chorioamnionitis [17], Prolonged Rupture of Membrane (PROM) [16], perinatal asphyxia a [16], positive blood culture [5], and group B streptococcus bacteremia [6].

In neonates, signs and symptoms of serious infections are often obscure, and the clinical presentation of neonatal meningitis is indistinguishable from that of neonatal sepsis without meningitis [5,6]. The most important clinical indicators of neonatal meningitis are convulsions, irritability, bulging fontanel, and temperature $\geq 39°C$ [11]. Therefore, the only confirmatory diagnosis for neonatal meningitis is cerebrospinal fluid analysis. The indications for obtaining cerebrospinal fluid for the diagnosis of meningitis in sepsis are controversial in most setups. Considering the difference in the magnitude of meningitis in EONS and LONS, routine CSF analysis is done in most neonatal units for all neonates with LONS whereas based on indications like positive neurologic symptoms, positive blood culture, and maternal chorioamnionitis in EONS [20]. However, to our knowledge, there is no data that studied the difference in magnitude to rationalize this clinical practice in most developing countries like Ethiopia.

Hence, the current study aimed to determine the magnitude, clinical presentations and associated risk factors of neonatal meningitis among newborns with neonatal sepsis.

## Methods and materials

### Study design, study area, and study period

An institution-based cross-sectional study was conducted among neonates with neonatal sepsis admitted to University of Gondar Comprehensive Specialized Hospital (UOGCSH) neonatal ward from May 1, 2021, to Oct 30, 2021. University of Gondar Comprehensive Specialized Hospital is one of the oldest tertiary teaching hospitals located in northwest Ethiopia. The neonatal ward in the hospital is a level III Neonatal Intensive Care Unit (NICU) that gives service to all neonates with medical and surgical illnesses.

Neonatal care is provided by consultant pediatricians, guest neonatologists, pediatric surgeons, pediatric residents, general practitioners, and neonatal nurses. The number of admissions ranges from 180–200 patients per month and an average of 3 cases of suspected neonatal sepsis are admitted per day. The commonest causes of admissions are neonatal sepsis, pneumonia, meningitis, prematurity, perinatal asphyxia, and neonatal jaundice.

### Study population

All neonates admitted with the diagnosis of neonatal sepsis during the study period.

### Eligibility criteria

**Inclusion criteria.** All neonates <28 days old with neonatal sepsis who had CSF analysis during the study period.

**Exclusion criteria.** Neonates with incomplete information were excluded.

### Sample size and sampling techniques

The sample size required in this study was calculated by using single population proportion formula, assuming the magnitude of neonatal meningitis of 17.9% from the study done in Kenya, using 95% CI (Z = 1.96), and 5% marginal error (w = 0.05). Since the source population was less than 10,000, a correction formula was applied. Using the above assumption and adding 5% contingency made the final sample size 185. The number of admitted newborns with neonatal sepsis in the study period from the previous admission-discharge registry was estimated to be 540. A systematic random sampling technique was used, and the interval k was 2.9, so we select every third newborn with neonatal sepsis from the sampling frame. The first study participant was the first MRN number on the first date of data collection.

### Data collection and procedure

Data was collected by using pretested structured questionnaires containing the socio-demographic characteristics of the mother& newborn, medical and obstetrics parameters of the mother, clinical presentation, and laboratory profile of the neonates. Pretest was done in 5% of the calculated sample size at the Debre Tabor Hospital, Ethiopia. Then, the necessary modifications were done based on the analysis. Two trained pediatric residents were used as data collectors. The principal investigators supervise the data collection daily and the study team were not involved in the diagnosis and treatment of the study participants. After interviewing the mother, the medical recordings of both the mother and the newborn were reviewed for further relevant information. The laboratory parameters including CSF analysis ordered by the treating physician were taken for laboratory analysis.

## Data compilation and analysis

The collected data was coded, checked for completeness, entered into Epi-info version 7.0, and exported to SPSS version 24.0 for analysis. Data was reported as the mean and standard deviation for continuous variables and percentages for categorical variables. Multivariable logistic regression method was used after conducting omnibus Tests of Model Coefficient, Hosmer and Lemeshow Test, and overall accuracy of the model. CI of 95% and p-value <0.05 was considered statistically significant.

## Operational definitions

**Sepsis**: is suspected or proven infection plus 2 or more of the following; temperature instability, tachycardia, hypotension, tachypnea, apnea, hypoxemia, renal failure, and leukocytosis/leukopenia [20].

 **Leukopenia**: WBC count < = 5000cell/mm3 [21,22].

 **Leukocytosis**: WBC count > = 20000 cell/mm3 [22].

 **Neonatal meningitis**: neonate with the feature of sepsis and positive CSF analysis [22,23].

 **Positive CSF for meningitis**: cell count >20cell/microliter and/or positive CSF culture, positive gram stain, CSF protein >150mg/dl preterm and >100mg/dl in term, glucose <35mg/dl for term and <25mg/dl for preterm or lower than 2/3rd of serum glucose [13,23].

 **Ethical considerations.** The ethical clearance was obtained from the ethical review board of the school of medicine, the University of Gondar in accordance with the declaration of Helsinki. The board provided an approval letter with the reference number ERB241/2021. Written informed consent was obtained from the parents of the study participants after the purpose of the study was explained. All participant parents were informed of their right to withdraw from the study at any stage of the research. All participants' personal information was kept confidential their name was not mentioned in the study.

## Results

### Socio-demographic characteristics of the study participants

Out of 185 randomly selected patients, 171 neonates were included in the study and, 14 were excluded (9 with incomplete documentation, 5 refused to participate). The male-to-female ratio of the participants was 1.3:1. The mean postnatal age of the study participants was 10.74 ±8.0 days. The majority (59.1%) of the newborns were greater than 7 days at presentation. Most of the newborns were born to mothers aged between 20 and 35 (**Table 1**).

### Maternal and neonatal health profiles

Among mothers who participated in this study, 163(95%) had ANC follow-up during pregnancy. Sixteen (10%) of the mother had a history of urinary tract infection during the third trimester. Fifteen (9%) of the mother were diagnosed with maternal chorioamnionitis at the time of delivery. Half (8) of the mothers with chorioamnionitis were given therapeutic antibiotics before delivery. About 24(14%) had PROM for more than 18 hours whereas 20(12%) had prolonged labor for more than 24 hours. The mean GA of the newborns was 37.4±2.38 weeks. The mean birth weight of the study participants was 2733.92±628.36 grams. Most (99%) newborns had normal APGAR scores at 5[th] minutes >7 (**Table 2**).

**Table 1. Socio-demographic characteristics of study participants (N = 171) admitted to UOGCSH, 2021.**

| Variables | Frequency | Percent |
|---|---|---|
| **Age of neonate** | | |
| <7 days | 70 | 40.9 |
| 7–28 days | 101 | 59.1 |
| **Sex** | | |
| Male | 97 | 56.7 |
| Female | 74 | 43.3 |
| **Maternal age (years)** | | |
| <20 | 8 | 4.68 |
| 20–35 | 159 | 92.98 |
| > = 35 | 4 | 2.34 |
| **Maternal Education** | | |
| Unable to write & read | 70 | 40.9 |
| Primary education | 40 | 23.4 |
| Secondary Education | 28 | 16.4 |
| College and above | 33 | 19.3 |
| **Family Income** | | |
| <2000ETB (<50$) | 84 | 49.1 |
| 2000-5000ETB | 68 | 39.8 |
| >5000ETB | 19 | 12 |
| **Residence** | | |
| Urban | 79 | 46.2 |
| Rural | 92 | 53.8 |

**Table 2. Maternal and neonatal health-related characteristics (N = 171) admitted to UOGCSH, 2021.**

| Variables | N (%) |
|---|---|
| **ANC follow-up** | |
| **Yes** | 163(95.3) |
| **No** | 8(4.7%) |
| **Maternal chorionamnionitis** | |
| **Yes** | 15(8.8) |
| **No** | 156(91.2) |
| **ROM** | |
| **<18 hours** | 147(86) |
| **> = 18 hours** | 24(14) |
| **Duration of labor** | |
| **<24hours** | 151(88.3) |
| **> = 24 hour** | 20(11.7 |
| **Maternal UTI** | |
| **Yes** | 16(9.4) |
| **NO** | 155(90.6) |
| **GA weeks** | |
| **< 37 weeks** | 68(39.8) |
| **≥37 weeks** | 103(60.20 |
| **Mean birth weight (gm.)** | |
| **< 2500gm** | 42(24.5) |
| **≥2500gm** | 129(75.5) |
| **APGAR score at 5th minute** | |
| ≥7 | 168(98.8) |
| <7 | 3(1.2) |
| **Antibiotic use prior to LP** | |
| Yes | 57(33.3) |
| No | 114(66.6) |

**Keys;** LP: Lumbar puncture, UTI: Urinary tract infections.

## Clinical and laboratory profiles of the study participants

The most common clinical features of neonatal meningitis in the study participants were failure to breastfeed 29(88%), fever 23(69.7%), vomiting 15(45.5%), seizure 7(21.2%), altered mentation 4(12.1%) and respiratory distress 8(33.3%) (**Table 3**).

Regarding the CSF analysis criteria used for the diagnosis of meningitis, 28(85%) were diagnosed with high cell count, 4(12.1%) with positive gram stain, and one patient had positive CSF culture Gram stain from CSF fluid was positive in 5(15.2%) of newborns with meningitis. Only 2(6%) patients with meningitis had isolated organisms (klebsiella species, and gram-negative rod). The mean CSF glucose was 44.5mg/dl (30–58.9) in those who had meningitis compared to 61.8mg/dl (44.7–78.9) in those without meningitis.

One-third (57) of the total patients had been treated with parenteral antibiotics before CSF was taken with the mean duration of 7.02 hours and two-thirds (22) of meningitis patients were also treated with parenteral antibiotics before the CSF sample was taken with the mean duration of 14 hours.

## Prevalence of neonatal meningitis

The overall prevalence of neonatal meningitis among the study participants was 19.3%; 95% CI [13.5%-25.1%]. The prevalence of meningitis was higher in EONS 22.8% (16/70) than in LONS, which was 16.8% (17/101) (**Fig 1**). The mean post-natal age for early-onset neonatal meningitis was 2.8 days whereas 19 days for late-onset meningitis.

## Factors associated with neonatal meningitis

After adjusting for confounders with the multivariable logistic regression model, prolonged ROM (AOR: 5:38; 95% CI: 1.92–14.42) and prolonged labor (AOR: 4.98; 95% CI: 1.99–12.48) were significantly associated with neonatal meningitis (**Table 4**).

## Discussion

Our study aimed to determine the prevalence of neonatal meningitis and found that the prevalence rate of 19.3% among the study participants. This finding is consistent with the studies done in other African countries like Kenya, (17.9%) [13], and Egypt, (20%) [12] but higher than the studies done at Addis Ababa, Ethiopia, (4.7%) [1], and Australia, (9.2%) [10]. This variation in prevalence may be due to the difference in socio-demographic characteristics of the study population, the sample size of the studies, and the criteria to diagnose meningitis. In

**Table 3. Clinical Manifestations of the study participants, and meningitis patients (N = 171) admitted to UOGCSH, 2021.**

| Variable | All<br>N (%) | Meningitis<br>N (%) |
|---|---|---|
| Failure to feed | 120(70.2) | 29(87.9) |
| Seizure | 14(8.2) | 7(21.1) |
| Altered mentation | 14(8.2) | 4(12.1) |
| Respiratory distress | 42(24.6) | 8(33.3) |
| Fever | 104(60.8) | 23(69.7) |
| Vomiting | 66(38.6) | 15(45.5) |

Most (94%) of the study participants had blood sample collected for culture and sensitivity analysis but only 7(4%) of the patients had positive results.

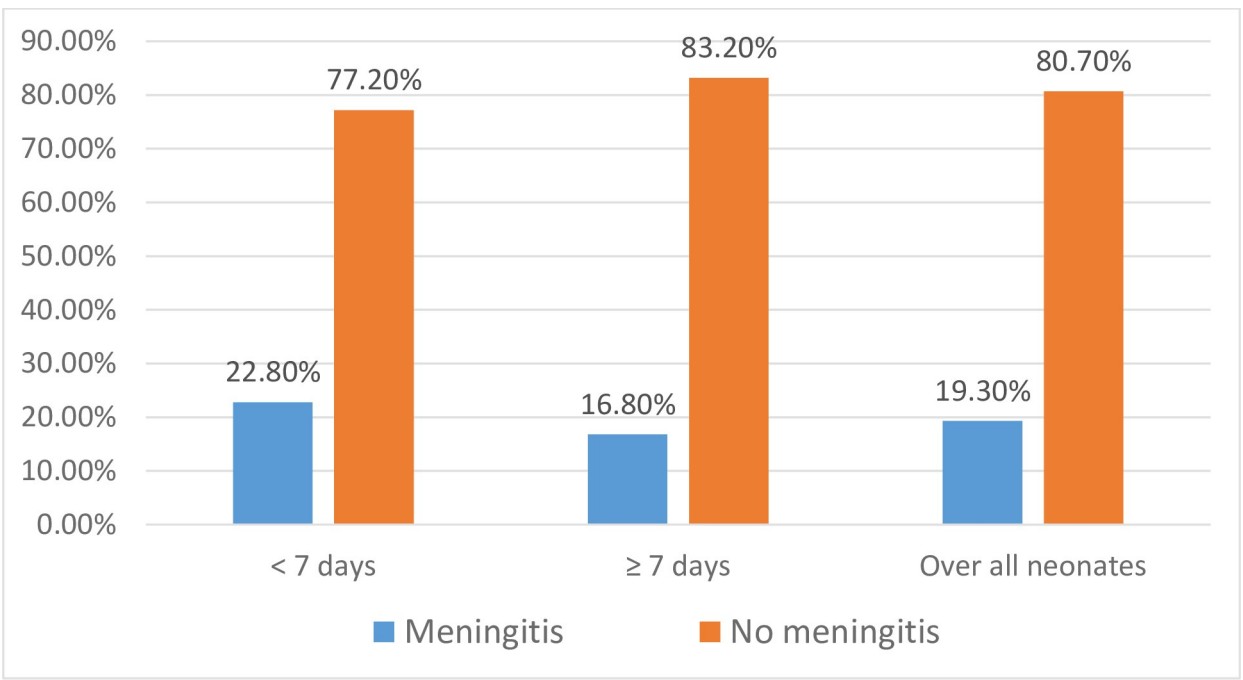

**Fig 1. Prevalence of neonatal meningitis among the study participants (N = 171) admitted to UOGCSH, 2021.**

the studies done in Addis Ababa, Ethiopia, and Australia, neonatal meningitis was diagnosed based on positive CSF culture result, which scientifically and practically does not show the real burden of meningitis in neonates. CSF culture is not 100% sensitive and it is affected by prior bactericidal antibiotic administration. Hence, using CSF culture as the only criteria for the diagnosis of meningitis might miss positive cases by other diagnostic tests. In our study, in addition to the CSF Culture, we used CSF cell count and gram stain which increased the yield for the diagnosis of neonatal meningitis.

**Table 4. Bi-variable and multivariable logistic regression of associated factors for neonatal meningitis among the study participants (N = 171) admitted to UOGCSH, 2021.**

| Variable | No meningitis | Meningitis | COR (95%, CI) | AOR | P - Value |
|---|---|---|---|---|---|
| | N (%) | N (%) | | (95%, CI) | |
| **GA** | | | | | |
| <37 weeks | 58(42) | 10(30.3) | 1.66(0.73-3.77) | 1.36(0.46-3.98) | 0.576 |
| >=37weeks | 80(58) | 23(69.7) | | | |
| **Birth weight** | | | | | |
| <2500gms | 35(25.4) | 7(21.2) | 1.26(0.504-3.162) | 0.79(0.23-2.68) | 0.715 |
| >=2500gms | 103(74.6) | 26(78.8) | | | |
| **DOL** | | | | | |
| <18 hrs. | 121(87.7) | 16(48.5) | | | |
| >=18hrs. | 17(12.3) | 17(51.5) | 5.56(2.08-14.86) | 4.98(1.99-12.48) | **0.001*** |
| **ROM** | | | | | |
| <18hrs. | 127(74.3) | 20(60.6) | | | |
| >=18 hrs. | 11(6.4) | 13(39.4) | 7.50(2.95-19.04) | 5.27(1.92-14.42) | **0.001*** |

**Notes:** * p-value < 0.05.

**Abbreviations:** COR = Crude Odds Ratio; AOR = Adjusted Odds Ratio.

In addition, the current study showed that CSF culture-confirmed meningitis was only 6%, which is nearly similar to the study from Kenya which was 4.8% [13] but lower than in other studies [16,24]. This lower rate of meningitis in our study might be due to antibiotics administration before the lumbar puncture in two-thirds of our study participants and the absence of rich culture media, especially for L. monocytogens. It is scientifically explained that a few doses of bactericidal antibiotics can result in the sterilization of the CSF finding making false negative culture and gram stain results. So in such scenarios, CSF cell count, latex agglutination, and CSF proteins are essential parameters for the diagnosis of partially treated meningitis [25].

The present study investigated that, the prevalence of meningitis was higher in early-onset neonatal sepsis 22.8% than in LONS, which was 16.8%. The current finding in this study was similar to the study done in Iran [16], among 20 meningitis cases, 65% had early onset meningitis (EOM); and in Taiwan [26], among 85, neonatal meningitis cases, 60% had EOM. In another studies, the prevalence of neonatal meningitis was higher in LONS than in EONS [10,27]. The dominance of early onset meningitis in our study might be due to the presence of study participants who had strong maternal risk factors (prolonged labor and prolonged premature rupture of the membrane) for early onset meningitis than LOM.

In the current study, two obstetric factors were strongly associated with neonatal meningitis. Newborns born to mothers with prolonged labor had five times odds of developing neonatal meningitis than newborns who had no prolonged labor, which is supported by the studies done in Iran [16] and Indonesia [28]. In addition, our study showed that prolonged PROM had more than five times the risk of having neonatal meningitis than neonates who did not have PROM, similar to a study in Indonesia where rupture of the membrane for more than 24 hours increased the risk of neonatal infection by 3 times [28]. This is due to prolonged PROM exposing the sterile intrauterine environment to pathogenic bacteria or the infant may be colonized while passing through the birth canal in the genitourinary tract. The common organisms that might benefit from PROM are GBS, Escherichia coli, Klebsiella, and L. monocytogens.

In our study, none of the neonatal factors were associated with neonatal meningitis. The male to female ratio amongst our study participants with meningitis was 1.3:1, similar well with other studies [13,29], which might be due to a sex-linked genetic susceptibility to meningitis in males [16,20].

In this study, neonatal meningitis's presenting signs and symptoms were nonspecific. The most common clinical presentations of the study participants with meningitis were failure to breastfeed, fever, and vomiting similar to other studies [13,16]. CNS manifestations are less common and were seen only in one-fourth of meningitis cases. The common neurologic symptoms seen in our patients were seizure and altered mentation which are also seen in other studies [12,16]. Therefore, the absences of neurologic symptoms are not a reliable indicator of the absence of neonatal meningitis in newborns with neonatal sepsis.

In the present study, blood culture was positive only in 6% of the meningitis cases, unlike other studies in which blood culture was a strong indicator of neonatal meningitis [10,30]. The lower rate of positive blood culture in our study might be due to the presence of a high number of participants who took antibiotics before lumbar puncture and low quality of laboratory in our hospital. So, relying on positive blood culture to obtain CSF sample in septic newborns will miss the majority of neonates with neonatal meningitis.

The limitation of the study was that we did not use the CSF protein to diagnose meningitis. The test was not available in our hospital during the study period.

## Conclusion and recommendation

The prevalence of neonatal meningitis was higher in EONS than in LONS. Prolonged labor and PROM were associated with neonatal meningitis. We recommend that routine CSF analysis should be practiced in all neonates with neonatal sepsis regardless of their postnatal age and clinical features. In addition, early diagnosis and treatment of obstetric complication may reduce neonatal meningitis. Further multicenter studies with rigorous methods are needed to develop better evidence.

## Supporting information

**S1 Checklist. STROBE statement—a checklist of items that should be included in reports of observational studies on.**
(DOC)

## Acknowledgments

The authors like to thank the study participants, data collectors, colleagues, the Department of Pediatrics and Child Health, and the University of Gondar.

## Author Contributions

**Conceptualization:** Mulugeta Nigusu Wondimu, Alemayehu Teklu Toni, Teshome Geletaw Zamanuel.

**Data curation:** Mulugeta Nigusu Wondimu, Alemayehu Teklu Toni, Teshome Geletaw Zamanuel.

**Formal analysis:** Mulugeta Nigusu Wondimu, Alemayehu Teklu Toni, Teshome Geletaw Zamanuel.

**Funding acquisition:** Mulugeta Nigusu Wondimu.

**Investigation:** Mulugeta Nigusu Wondimu, Alemayehu Teklu Toni, Teshome Geletaw Zamanuel.

**Methodology:** Mulugeta Nigusu Wondimu, Alemayehu Teklu Toni, Teshome Geletaw Zamanuel.

**Project administration:** Mulugeta Nigusu Wondimu, Teshome Geletaw Zamanuel.

**Resources:** Mulugeta Nigusu Wondimu, Alemayehu Teklu Toni, Teshome Geletaw Zamanuel.

**Software:** Mulugeta Nigusu Wondimu, Alemayehu Teklu Toni, Teshome Geletaw Zamanuel.

**Supervision:** Mulugeta Nigusu Wondimu, Alemayehu Teklu Toni, Teshome Geletaw Zamanuel.

**Validation:** Mulugeta Nigusu Wondimu, Alemayehu Teklu Toni.

**Visualization:** Mulugeta Nigusu Wondimu, Alemayehu Teklu Toni, Teshome Geletaw Zamanuel.

**Writing – original draft:** Mulugeta Nigusu Wondimu, Teshome Geletaw Zamanuel.

**Writing – review & editing:** Alemayehu Teklu Toni, Teshome Geletaw Zamanuel.

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
