## [Decision Letter · Decision Letter 0]

6 Apr 2023

PONE-D-22-30428Higher prevalence of neonatal meningitis in early-onset sepsis than late-onset among newborns admitted to the University of Gondar Comprehensive Specialized Hospital, Ethiopia.PLOS ONE

Dear Dr. Teshome Geletaw Zamanuel

Thank you for submitting your manuscript to PLOS ONE. After careful consideration, we feel that it has merit but does not fully meet PLOS ONE’s publication criteria as it currently stands. Therefore, we invite you to submit a revised version of the manuscript that addresses the points raised during the review process.

We look forward to receiving your revised manuscript.

Kind regards,

Abera Mersha, MSc.

Academic Editor

PLOS ONE

Journal Requirements:

Reviewers' comments:

Reviewer's Responses to Questions

**Comments to the Author**

1. Is the manuscript technically sound, and do the data support the conclusions?

Reviewer #1: Yes

Reviewer #2: Partly

2. Has the statistical analysis been performed appropriately and rigorously? 

Reviewer #1: Yes

Reviewer #2: No

3. Have the authors made all data underlying the findings in their manuscript fully available?

Reviewer #1: Yes

Reviewer #2: No

4. Is the manuscript presented in an intelligible fashion and written in standard English?

Reviewer #1: Yes

Reviewer #2: No

5. Review Comments to the Author

Reviewer #1: Manuscript ID: PONE-D-22-30428

Manuscript Title: Higher prevalence of neonatal meningitis in early-onset sepsis than late-onset among newborns admitted to the University of Gondar Comprehensive Specialized Hospital, Ethiopia

General

1. Is the manuscript technically sound, and do the data support the conclusions?

Yes, the manuscript technically sound that has used data to support the conclusion

2. Has the statistical analysis been performed appropriately and rigorously?

Yes, but the regression models can be reduced by deleting non-significant data see table 4

3. Have the authors made all data underlying the findings in their manuscript fully available?

Yes, it is available

4. Is the manuscript presented in an intelligible fashion and written in standard English?

Yes, it is written in standard English but needs corrections and editions throughout the process

5. Review Comments to the Author

Line 1-3, Title: I suggest to modify the title as prevalence of neonatal meningitis in early-onset sepsis among newborns admitted to the University of Gondar Comprehensive Specialized Hospital, Ethiopia: cross sectional survey

Line 39, Keywords: add few words like neonatal meningitis

Abstract: avoid abbreviations from the abstract see line 36-37 of the abstract e.g., EONS, LONS and ROM

Introduction

Either wright fully for the first time or do not use too much abbreviations’ in the txt see line 60 GBS and 66 CSF etc.

Methods:

see line 77; UOGCSH, NICU

Sample size and sampling techniques

Line 94-100: Sample size calculation

The single population proportion formula is ok but the sample size failed to consider correction formula because the source population were 180-200 or 540 as you indicated on line 100, which is less than 10,000.

Sampling methods

Line 100: Was it convenient to select every third from the total population of 540

What was the problem when you excluded the incomplete documentation

Line 103: Data collection and procedure

Line 104: It would be nice if you indicate where the pretest was conducted?

Line 127: Results

The calculated value was 185 and the response rate became 171(92%) of the total population. The contingency added in the calculated sample size was 5%; However, the response rate is 92% which is mor than the contingency added. what is the justification for this problem?

Table 1: Socio demographic factors: Family income in this result is only 103, what is the justification and what is the income of 68 individuals?

Table 2: Maternal and neonatal health-related characteristics

ROM total population is 162 justifications for this and the loss of 9 participants

Table 3: Clinical Manifestations of the study participants

This table is not clear either redefine or make it clear

Figure 1 needed to add percentages on the top of each bar

Table 4: Factors associated with neonatal meningitis:

GA Row 1 is not significant and Birth weight is not significant are not significant is so we do not have to overcrowd the table

Line 184-240: Discussion: Minimize the discussion and focus on the main points

Line 157: Ethical considerations

What do you mean by “Written informed consent was obtained from the parents of the study participants after the purpose of the study was explained.”

If you did written meaning did you take signature of the participants? If so, how did you manage those who do not read and write? Or did you take finger print? It is important to clarify this issue

Reviewer #2: Title: Higher prevalence of neonatal meningitis in early-onset sepsis than late-onset among newborns admitted to the University of Gondar Comprehensive Specialized Hospital, Ethiopia

Thank you for considering me to review this manuscript. The paper raises one of the main causes of neonatal morbidity and mortality, particularly in developing nations including Ethiopia. I put my concerns as follows:

Title

#1. The title includes unnecessary phrases or information; the study about the prevalence and associated factors of neonatal meningitis and the onset of sepsis can be treated as one of the factors. Why are you interested in giving special emphasis to it? Even in the result section, only the proportion difference (22.8% (EONS) vs 16.8% (LONS)) was used to say higher rather it must be supported with statistical tests like proportion (mean) difference with the p-value. In short, I suggest the authors modify the title to, “Prevalence of neonatal meningitis and associated factors among newborns admitted to the University of Gondar Comprehensive Specialized Hospital, Ethiopia”

#2. The target population must be consistent throughout the document. Eg. In the title, your target populations are “newborns admitted to the University of Gondar Comprehensive Specialized Hospital”. When you go down to the abstract the target population is “newborns with neonatal sepsis”, and “newborns with suspected or confirmed sepsis”. In the method section, “All neonates admitted with clinically suspected sepsis”.

Abstract

#1. Remove all the abbreviations.

#2. In the method section, key information such as sample size, sampling technique, and software used for data entry are missed

#3. When you report the prevalence of neonatal meningitis, you include the 95% confidence interval. #4. The conclusion is almost a copy of the result section and it lacks recommendations.

Introduction

#1. The introduction does not tell us why this study was conducted. You tried to mention some information starting from lines 70 & 71 as “to our knowledge, there is no data that studied the difference in magnitude to rationalize this clinical practice in most developing countries like Ethiopia” but as you cited in Ethiopia there is “a 10-year retrospective review on Neonatal meningitis”. What new information do we get from this study?

#2. The introduction fails to address previously proposed national or local solutions to minimize the morbidity and mortality associated with the problem under study (neonatal meningitis).

Methods

#1. There are some contradictory ideas in the method section. In line 76, you write as the study is prospective on the other hand in the exclusion criteria (line 92) you exclude cards with incomplete documentation. One advantage of the prospective study is to overcome such problems so why do you exclude them? In addition, you exclude “neonates whose families or legal guardians refused to participate”, which is not the right approach rather you have to consider those as non-responders.

#2. Sample size. First, why did you take the proportion from Kenya despite there being a study in Ethiopia? Second, in such rare proportions, as a right thumb rule, we have to decrease the margin of error or determine the sample size using the factors to get an appropriate sample size. In this study, the sample size is 185, which is not enough for logistic regression. Even when you look at the regression table, the confidence intervals are too wide, which may be due to the small sample size. In contrast, in the last paragraph of the discussion, you mention using an adequate sample size as one of the strengths of this study. Do you think the sample size is adequate? I understand you take over 6 months even to get this sample but there are options to get an adequate sample even with less period. eg. By including other health facilities. Furthermore, in line 82, you mentioned on average 3 cases of suspected neonatal sepsis are admitted per day” If so, why did you take six months (May 1, 2021, to Oct 30, 202 to collect data from 171 neonates even less than 1 per day.

#3. The data collection method is poorly written. First, there is no clear information about the data collection tool: validity, reliability, means of diagnosis… Second, how many data collectors were involved? Their experience. Which data collection method was used, interviewer-administered, chart review, or both? Based on the information written under the “Data collection and procedure” sub-section the main information source were charts of both the mother and the newborn, and CSF analysis ordered by the treating physician. If that is the case, why do you make the study perspective? Is there any primary information collected either from the mother or newborn for this study only?

#4. Replace multivariate with multivariable; there is a big difference between them.

#5. I have doubts about your operational definition. If the newborn is admitted to NICU with suspected sepsis empirical management using antibiotics may be initiated, and the CSF analysis is sent if they are unresponsive to the medication. CSF analysis after the initiation of antibiotics is not accurate and has a significant effect on the overall prevalence of meningitis. What is your justification for this?

Result

#1. In line 143, most (99%) newborns had normal Apgar 144 scores>7. First, which minute Apgar score is it? Second, do you think newborns with suspected sepsis are presented with such an Apgar score?

#2. It is unclear how did you assess some variables such as maternal chorioamnionitis, Maternal UTI, ROM, and even GA. To get reliable information about them the mother must deliver them to the same hospital where the study was conducted. However, there may be referral cases, so how could you get such information? Was it self-report?

#3. All the information written under the result section must be based on the data collection tool and procedure mentioned under the method section. You report the blood culture and sensitivity results despite nothing being said in the method section. The detail of the CSF analysis is also missed in the method despite your report in the result section (i.e. “28(85%) was diagnosed with cell count, 4(12.1%) with gram stain, and one case by CSF culture”)

#4. The 95% CI is mandatory for your discussion. In addition, you write as “meningitis was higher in EONS 22.8% than in LONS, which was 16.8%”. I think, you just simply look at the proportion, 22.8%> 16.8, to say higher rather than prove it, is there a statistical difference between early and late-onset neonatal sepsis to encounter meningitis?

#5. The final model must be computed after including sepsis (early vs late) as one factor.

#6. The discussion, conclusion, and recommendation must be modified after updating the model.

6. PLOS authors have the option to publish the peer review history of their article (what does this mean?). If published, this will include your full peer review and any attached files.

Reviewer #1: No

Reviewer #2: No

---

## [Author Response · Author response to Decision Letter 0]

20 May 2023

To the editors

We acknowledge the editors for providing us plausible comments. We have corrected accordingly.

To the Reviewer 1

We really thank you for giving us the informative comments. We have corrected based on the comments.

To the reviewer 2

We thank you for the comments. But We are a bit disappointed for giving us “No“ for all the first four questions while the first reviewer provided us “yes“ for all of the first four questions. However the specific comments given in the manuscript are constructive and meaningful.

---

## [Decision Letter · Decision Letter 1]

18 Jun 2023

PONE-D-22-30428R1Magnitude of Neonatal Meningitis and Associated Factors Among Newborns with Neonatal Sepsis Admitted to the University of Gondar Comprehensive Specialized Hospital, North Gondar, Ethiopia.PLOS ONE

Dear Dr. Zamanuel,

Thank you for submitting your manuscript to PLOS ONE. After careful consideration, we feel that it has merit but does not fully meet PLOS ONE’s publication criteria as it currently stands. Therefore, we invite you to submit a revised version of the manuscript that addresses the points raised during the review process.

We look forward to receiving your revised manuscript.

Kind regards,

Abera Mersha, MSc.

Academic Editor

PLOS ONE

Reviewers' comments:

Reviewer's Responses to Questions

**Comments to the Author**

1. If the authors have adequately addressed your comments raised in a previous round of review and you feel that this manuscript is now acceptable for publication, you may indicate that here to bypass the “Comments to the Author” section, enter your conflict of interest statement in the “Confidential to Editor” section, and submit your "Accept" recommendation.

Reviewer #1: All comments have been addressed

Reviewer #2: All comments have been addressed

2. Is the manuscript technically sound, and do the data support the conclusions?

Reviewer #1: Yes

Reviewer #2: Yes

3. Has the statistical analysis been performed appropriately and rigorously? 

Reviewer #1: Yes

Reviewer #2: No

4. Have the authors made all data underlying the findings in their manuscript fully available?

Reviewer #1: Yes

Reviewer #2: Yes

5. Is the manuscript presented in an intelligible fashion and written in standard English?

Reviewer #1: Yes

Reviewer #2: No

6. Review Comments to the Author

Reviewer #1: All comments are incorporated and corrected from my side and I have seen what the authors responded to my previous concern and comments. Thus, I accept to proceed to the next step

Reviewer #2: The majority of the concerns raised in the previous review were addressed but still the justification given for the final model needs further clarification. In addition, why do you only display significantly associated variables in table 4? Display all the variables included in the model with their respective measure of effects both in the bivariable and multivariable column.

7. PLOS authors have the option to publish the peer review history of their article (what does this mean?). If published, this will include your full peer review and any attached files.

Reviewer #1: **Yes: **Hussen Mekonnen Asfaw

Reviewer #2: No

---

## [Author Response · Author response to Decision Letter 1]

4 Aug 2023

We thank the editors and reviewers so much. They provided us tangible comments.

---

## [Decision Letter · Decision Letter 2]

13 Aug 2023

Magnitude of Neonatal Meningitis and Associated Factors Among Newborns with Neonatal Sepsis Admitted to the University of Gondar Comprehensive Specialized Hospital, North Gondar, Ethiopia.

PONE-D-22-30428R2

Dear Dr. Zamanuel,

We’re pleased to inform you that your manuscript has been judged scientifically suitable for publication and will be formally accepted for publication once it meets all outstanding technical requirements.

Kind regards,

Abera Mersha, MSc.

Academic Editor

PLOS ONE

Additional Editor Comments (optional):

Reviewers' comments:

Reviewer's Responses to Questions

**Comments to the Author**

1. If the authors have adequately addressed your comments raised in a previous round of review and you feel that this manuscript is now acceptable for publication, you may indicate that here to bypass the “Comments to the Author” section, enter your conflict of interest statement in the “Confidential to Editor” section, and submit your "Accept" recommendation.

Reviewer #2: All comments have been addressed

2. Is the manuscript technically sound, and do the data support the conclusions?

Reviewer #2: Yes

3. Has the statistical analysis been performed appropriately and rigorously? 

Reviewer #2: Yes

4. Have the authors made all data underlying the findings in their manuscript fully available?

Reviewer #2: Yes

5. Is the manuscript presented in an intelligible fashion and written in standard English?

Reviewer #2: Yes

6. Review Comments to the Author

Reviewer #2: (No Response)

7. PLOS authors have the option to publish the peer review history of their article (what does this mean?). If published, this will include your full peer review and any attached files.

Reviewer #2: No

---

## [Editor Report · Acceptance letter]

4 Sep 2023

PONE-D-22-30428R2 

Magnitude of Neonatal Meningitis and Associated Factors Among Newborns with Neonatal Sepsis Admitted to the University of Gondar Comprehensive Specialized Hospital, North Gondar, Ethiopia. 

Dear Dr. Zamanuel:

I'm pleased to inform you that your manuscript has been deemed suitable for publication in PLOS ONE. Congratulations! Your manuscript is now with our production department. 

Kind regards, 

on behalf of

Mr. Abera Mersha 

Academic Editor

PLOS ONE